# Marine-Derived Omega-3 Polyunsaturated Fatty Acids and Heart Failure: Current Understanding for Basic to Clinical Relevance

**DOI:** 10.3390/ijms20164025

**Published:** 2019-08-18

**Authors:** Atsushi Sakamoto, Masao Saotome, Keisuke Iguchi, Yuichiro Maekawa

**Affiliations:** Division of Cardiology, Internal Medicine III, Hamamatsu University School of Medicine, Hamamatsu 431-3192, Japan

**Keywords:** omega 3 polyunsaturated fatty acid, n-3 polyunsaturated fatty acid, eicosapentaenoic acid, docosahexaenoic acid, heart failure, cardiovascular disease

## Abstract

Heart failure (HF) is a rapidly growing global public health problem. Since HF results in high mortality and re-hospitalization, new effective treatments are desired. Although it remains controversial, omega 3 polyunsaturated fatty acids (n-3 PUFAs), such as the eicosapentaenoic acid and docosahexaenoic acid, have been widely recognized to have benefits for HF. In a large-scale clinical trial regarding secondary prevention of HF by n-3 PUFA (GISSI-HF trial), the supplementation of n-3 PUFA significantly reduced cardiovascular mortality and hospitalization. Other small clinical studies proposed that n-3 PUFA potentially suppresses the ventricular remodeling and myocardial fibrosis, which thereby improves the ventricular systolic and diastolic function both in ischemic and non-ischemic HF. Basic investigations have further supported our understanding regarding the cardioprotective mechanisms of n-3 PUFA against HF. In these reports, n-3 PUFA has protected hearts through (1) anti-inflammatory effects, (2) intervention of cardiac energy metabolism, (3) modification of cardiac ion channels, (4) improvement of vascular endothelial response, and (5) modulation of autonomic nervous system activity. To clarify the pros and cons of n-3 PUFA on HF, we summarized recent evidence regarding the beneficial effects of n-3 PUFA on HF both from the clinical and basic studies.

## 1. Introduction

Heart failure (HF) is a rapidly growing global public health problem both in developed and developing countries, with an estimated prevalence of over 37.7 million patients worldwide [1]. Despite recent developments of HF treatments, including pharmacologic and device therapy, HF results in high mortality and re-hospitalization. Therefore, innovations regarding HF treatments are desired.

Epidemiologic discovery first indicated that larger fish consumption afforded beneficial effects on cardiovascular disease (CVD). Observational and interventional studies, which were conducted from the 1980s to the early 2000s, indicated that marine-derived omega 3 polyunsaturated fatty acids (n-3 PUFAs, e.g., eicosapentaenoic acid (EPA) and docosahexaenoic acid (DHA)) are effective for the ischemic heart disease (IHD) [2,3], atrial [4] and ventricular arrhythmia [5], hypertension [6], peripheral artery disease [7], and HF [8,9,10]. In addition, large scale clinical trials further revealed that pharmacological intervention by purified n-3 PUFA significantly reduced the mortality of CVD [3,11]. In contrast to these studies, a number of clinical trials have shown that n-3 PUFA intervention provides less beneficial cardiovascular outcomes [12,13,14,15], posing a question regarding the true value of n-3 PUFA in the current status, where strong lipid-lowering management, including statins and PCSK-9 inhibitors, can be used for CVD patients. However, a recent large-scale clinical trial (REDUCE-IT; Reduction of Cardiovascular Events with Icosapent Ethyl–Intervention Trial) showed that a relatively high amount of n-3 PUFA supplementation (4 g/day) significantly reduced CVD events in patients with elevated serum triglyceride [16], and the beneficial effect of n-3 PUFAs intervention against CVD has been re-recognized.

Since multiple factors, such as the quality and quantity of n-3 PUFA supplementation and cohort profile (e.g., age, race, gender and educational status), can affect clinical outcomes, it is difficult to accurately assess the beneficial effects of n-3 PUFA on a heterogenous clinical syndrome like HF. Furthermore, due to the multiple pleiotropic pharmacological effects of n-3 PUFAs in vivo, a complete understanding of the pathophysiological mechanisms of n-3 PUFA on HF remains elusive. In the current review, we overview results of clinical trials with regard to both dietary and therapeutic intervention of n-3 PUFA for the purpose of cardio-protection, mainly focusing on HF, and discuss the underlying basic molecular mechanism of these effects.

## 2. Structure and Metabolism of PUFA

According to the definition, PUFAs are differentiated from saturated and monounsaturated fatty acids (FAs) by the presence of two or more carbon-to-carbon double bonds within their molecular structure. The location of the first double bond from the methyl (-CH_3_) or omega (n-) end of the chain provides the name of the PUFA, and n-3 PUFAs have a double bond at the site of the third carbon, whereas omega 6 PUFAs (n-6 PUFAs) have the bond at the sixth carbon from the methyl end of the chain (Figure 1). Since humans do not possess the desaturation enzyme synthesizing the carbon double bond after the ninth carbon from the methyl end, n-3 PUFAs are considered as “essential fatty acids”, meaning that they must be obtained from diet. The most common dietary n-3 PUFAs include EPA (20:5n-3) and DHA (22:6n-3), which originate primarily in fish oils, and alpha-linolenic acid (ALA; 18:3n-3), found in flaxseed, perilla, or camelina oils. Meanwhile, common n-6 PUFAs, arachidonic acid (AA; 20:4n-6) and linoleic acid (LA; 18:2n-6), are contained in animal fat and safflower oil, respectively. EPA and DHA can be converted from ALA, and, similarly, LA is metabolized to AA via an enzymatic reaction of delta-6 desaturase in vivo, although the efficacy is limited and altered by age, gender, and individuals [17]. Thus, the oral intake of EPA and DHA is crucial to maintain health benefits.

FAs are transported to the liver and several tissues after being absorbed at the intestinal tract as chylomicrons. During blood circulation, FAs exist as triglycerides or phospholipids involved in lipoproteins (e.g., very low-density and low-density lipoproteins), or free fatty acids binding to the serum albumin. FAs are subsequently incorporated into the cellular phospholipid bilayer of the plasma cellular membrane of whole-body tissues. FAs can also provide large quantities of intracellular ATP, through β-oxidation and the Krebs cycle in mitochondria. In fact, the heart requires FAs as an essential energy source, which provides about 70% of the total amount of energy under normal conditions. Of note, it is well known that the bioactive eicosanoids derived from PUFA have different reactions in terms of pro- or anti-inflammation, thrombotic activity, and vasocontractile or vasodilate reaction, depending on their parent PUFA compounds [18].

## 3. Clinical Evidence of n-3 PUFA on Heart Failure

HF is a condition in which the heart is unable to pump enough blood to meet demand. Since HF is a multifactorial syndrome, its mortalities and progressions differ from their underling etiologies, such as IHD, valvular heart disease, hypertension, arrhythmia, cardiac myopathies, congenital heart disease, endocrine and metabolic diseases, infection, and certain drugs. Several prior prospective observational studies and randomized control trials proved that consumption of fish or fish oil containing n-3 PUFA decreased IHD mortality (e.g., myocardial infarction (MI) and sudden cardiac death) in patients with or without pre-diagnosed CVD [2,19,20]. In addition, recent studies have shown that fish and/or n-3 PUFA intake also prevented the new onset of HF and rehospitalization [9,10,21,22,23,24]. However, conflicting results were reported from other groups [25,26,27], meaning that the n-3 PUFA-mediated effects on HF are once again viewed as uncertain. In this section, we summarize existing clinical evidence regarding primary and secondary prevention of HF, and discuss the considerable points which may cause such heterogenous results.

### 3.1. Primary Prevention of HF by n-3 PUFA

To date, a number of population-based prospective observational studies were conducted to reveal the effect of fish and fish oil for primary prevention of HF (Table 1), and they primarily used food-frequency questionnaires (FFQs) to estimate fish intake and subsequent calculation of the amount of food containing n-3 PUFA. In 2005, the Cardiovascular Health Study, which studied 4738 elderly US patients for 12 years, reported that fish consumption (broiled or baked, but not fried, tuna or other fish) was associated with a lower incidence of HF [8]. The following two large-scale studies in the US (Woman’s Health Initiative and Physicians’ Health study) exhibited an inverse correlation between fish consumption and HF incidence [9,22], and a Japanese large cohort study revealed that total fish consumption including “Tempura fried fish” and regular n-3 PUFA intake suppressed HF mortality [23]. Conversely, other large scale cohort studies performed in European countries (Rotterdam study and Swedish cohorts) failed to show supportive results regarding the primary prevention of HF by fish intake [25,26,27]. These conflicting results may be derived from diversities, such as genetic background for bioavailability of n-3 PUFA and HF phenotypes. The contamination of pollutants (e.g., methyl mercury, polychlorinated biphenyls (PCBs), and dioxins) in fish is reported to increase cardiovascular risk [19]. A recent study from a Swedish cohort revealed that the dietary intake of PCBs increased the risk of HF (up to 42%–48%), although the intake of EPA-DHA was associated with 18–29% lower risk of HF in mutually-adjusted models contrasting extreme quintiles of exposures [28].

### 3.2. Secondary Prevention of HF by n-3 PUFA

Several clinical studies were conducted to elucidate the effect of n-3 PUFA on secondary prevention for HF (Table 2). The GISSI-HF (Gruppo Italiano per lo Studio della Sopravvivenza nell’Infarto Miocardico–Heart Failure) trial was the first and is currently the only large-scale double-blind placebo-controlled RCT, which evaluated the effect of n-3 PUFA in patients with HF [10]. Among 6875 participants involving both ischemic and non-ischemic HF, 91% had HF with a reduced left ventricular ejection fraction (LVEF) (<40%). After follow up to 3.9 years, 1g daily n-3 PUFA supplementation reduced the risk of total mortality by 9% (adjusted hazard ratio (HR) 0.91 (95% CI 0.833–0.998), *p* = 0.041) and the risk of hospitalization due to cardiovascular reasons by 8% (adjusted HR 0.92 (99% CI 0.849–0.999), *p* = 0.009) [10]. Additionally, the reduction of the composite endpoint including all-cause death or admission to hospital for cardiovascular reasons appeared to be more pronounced in patients with EF ≤ 40% (HR 0.94 (0.88–0.99)) than EF > 40% (HR 1.02 (0.83–1.25)). In the echocardiographic sub-study of GISSHI-HF, n-3 PUFA treatment provided a small but statistically significant improvement regarding LVEF [30]. Although the extent of the benefit of n-3 PUFA was not high, the high prescription rate of other anti-HF agents in the participants, i.e., angiotensin converting enzyme inhibitor (ACE-I)/angiotensin receptor blocker (ARB) (93%) and β-blocker (BB) (64%), these results allow us to realize the significance of the n-3 PUFA intake in HF patients.

Small-scale clinical trials among ischemic and/or non-ischemic HF patients, which studied the improvement of cardiac function mainly on the basis of echocardiography, have been reported [24,31,32,33,34,35,36,37,38]. Nodari et al., in their double-blind RCT, revealed that a n-3 PUFA intake (2 g/day) improved the LVEF and HF symptoms in 133 patients with dilated cardiomyopathy (mean LVEF = 36%) [24]. During the 12 month follow up period, the n-3 PUFA group exhibited an increase in LVEF (by 10.4%), while the placebo group exhibited a decrease in LVEF (by 5.0%, *p* < 0.001), and showed improvement in the peak VO_2_, exercise duration, and New York Heart Association (NYHA) functional class. In addition, HF hospitalization was significantly suppressed in the n-3 PUFA group (6% vs 30% in the placebo group, *p* = 0.0002) [24]. Dose-dependent beneficial effects of n-3 PUFA supplementation was also reported by Moertl et al. in their non-ischemic cardiomyopathy patient cohort [33]. In this RCT, 1 to 4 g/day of n-3 PUFA (EPA and DHA) treatment significantly improved LVEF in a dose-dependent fashion [33].

The Omega-3 Acid Ethyl Esters on the Left Ventricular Remodeling After Acute Myocardial Infarction (OMEGA-REMODEL) RCT, which was conducted with 358 patients with acute MI [36], revealed that an additional n-3 PUFA intake (4 g/day for six months) with ordinal current treatment (the guideline-recommended therapies against LV remodeling after MI, e.g., ACE-I/ARB, BB, and emergent percutaneous coronary revascularization) suppressed adverse LV remodeling after MI. In cardiac magnetic resonance imaging, the LV end-systolic volume index (LVESVI: −5.8%, *p* = 0.017) and the non-infarct myocardial fibrosis (−5.6%, *p* = 0.026) were significantly suppressed in the n-3 PUFA group compared to those in the placebo group. The serum biomarkers regarding systemic and vascular inflammation were significantly suppressed, which suggested that the suppression of myocardial fibrosis would be one of the mechanisms of n-3 PUFA-mediated protection on damaged myocardium. Cardiovascular event reduction in the GISSI-HF trial was mainly obtained by anti-arrhythmic effects [10], and small-scale clinical trials also supported anti-arrhythmic effects by n-3 PUFA in patients with HF [39,40]. These data suggest that the anti-arrhythmic effects of n-3 PUFA may contribute to CVD events in patients with HF.

To date, the European Society of Cardiology guidelines give a class IIb recommendation (with level B evidence) regarding n-3 PUFA treatment for symptomatic HF patients to reduce the risk of cardiovascular hospitalization and cardiovascular death [41]. The American Heart Association advisory board proposed a stronger recommendation, in which n-3 PUFA treatment in patients with HF with reduced ejection fraction (HFrEF) is recommended as class IIa and level of evidence B [29]. Since HF is a heterogeneous and multifactorial syndrome, it is difficult to interpret the results from the clinical trials of n-3 PUFA on HF treatment. Further investigations regarding n-3 PUFA are required to explore the most effective phenotype and/or etiology of HF, and the appropriate quality and/or quantity of n-3 PUFA to obtain the most beneficial effects for the prevention of HF.

The effects of supplemental n-3 PUFA for the prevention of HF-related cardiac cachexia must also be clarified. A small randomized double-blind trial, which was conducted in a placebo-controlled study with 14 symptomatic HF patients (NYHA III or IV), revealed that a high-dose n-3 PUFA (8 g/day) intake for 18 weeks significantly decreased the lipopolysaccharide-induced tumor necrosis factor-alpha (TNF-α) released from patients’ peripheral blood mononuclear cells, as well as improving body weight, suggesting the potential benefit of n-3 PUFA against cardiac cachexia [32]. Further larger scale clinical trials are desired to ensure the effects of n-3 PUFA on the prevention of HF-related cachexia.

Although it is well-known that HF preserved ejection fraction (HFpEF) accounts for up to half of all HF in developed nations [42], to the best of our knowledge, there are fewer published reports which investigate the benefits of n-3 PUFA on patients with HFpEF. Since n-3 PUFA can ameliorate interstitial fibrosis, which is a well-known characteristic of the HFpEF heart [36,38], it would be effective to suppress the pathogenesis and/or progression of HFpEF as well as HFrEF.

### 3.3. Key Points Which Can Affect Heterogeneous CVD Outcomes

As mentioned above, heterogeneous CVD outcomes have been reported from PUFA clinical trials. These heterogenous outcomes in CVD may be derived from differences in the quality and quantity of n-3 PUFAs adopted in each clinical study, and other baseline clinical background factors, including genetic variance in terms of n-3 PUFA bioavailability. In this section we discuss the key points which can affect the results of PUFA clinical trials.

Heterogeneity in the blood and tissue PUFA concentration: Although a previous meta-analysis reported no statistically significant relationship between n-3 PUFA consumption and CVD mortality, the variability in individual serum n-3 PUFA levels was not taken into consideration in the study [43]. Superko et al. reported that a fixed-dose n-3 PUFA administration exhibited significant individual variation in the blood n-3 PUFA levels [44], whereas, in general, dietary intake of n-3 PUFA is closely associated with its blood levels [45,46]. It is well known that the risk reduction of CVD depends on n-3 PUFA blood concentration [44], and an inverse correlation with the blood n-3 PUFA profiles has been reported in IHD [47,48], arrhythmia [49], peripheral artery disease [50], and HF [21,51]. Thus, heterogenous tissue and/or blood concentration in each patient (even for the same dose intake) may affect heterogenous CVD outcomes.

Effective dose and bioavailability of PUFA: Currently, few studies have provided an ideal n-3 PUFA dose, which can sufficiently correct the tissue PUFA concentration, and which may account for the heterogeneity of clinical outcomes. This unavailability of an effective dose of PUFA intake for HF prevention decreases the clinical evidence level. Thus, in order to use PUFA for HF prevention, we should first explore titration of the ideal n-3 PUFA dose by monitoring blood and/or tissue PUFA concentrations. In addition, the bioavailability of n-3 PUFA can be altered by its chemical form. In triglyceride or phospholipid forms (as usually found in fish oil), n-3 PUFA is mainly hydrolyzed by colipase-dependent pancreatic lipase. Meanwhile the bioavailability of the ethyl-ester form of n-3 PUFA is highly dependent on the presence of dietary fat, which can stimulate the release of bile salts into the small intestinal tract, because n-3 PUFA ethyl-esters requires digestion with bile salt-dependent lipase [52,53]. Currently, the US Food and Drug Administration approves two ethyl-ester (Lovaza^®^, GlaxoSmithKline (EPA/DHA ethyl-esters) and Vascepa^®^, Amarin Pharmaceuticals (EPA ethyl-esters)) and one free fatty acid (Epanova^®^, AstraZeneca (EPA/DHA free fatty acids)) forms of n-3 PUFA supplement. Thus, the different chemical forms and bioavailability of n-3 PUFA can affect the clinical outcome through absorption via the intestine. Regarding bioavailability, the free forms of n-3 PUFA seem to provide better CVD outcomes than the ethyl-ester forms [54,55]. Therefore, further clinical trials to investigate differences of bioavailability are required.

Genetic variant in PUFA metabolism: It is likely that genetic polymorphism affects individual taste-perception, which decisively determines the intake of food containing n-3 PUFA (e.g., oily fish). Prior familial aggregation analysis indicated the strong heritability of FA composition in the Red blood cells (RBC) membrane (up to 40%–70%) [56]. The endogenous synthesis of n-3 PUFAs from plant-derived ALA appears mainly in the liver via desaturation and elongation reactions, even though the estimated conversion rate is low: 5% to 10% for EPA, and 2% to 5% for DHA [17]. Genetic variants of the key enzymes in this pathway, including delta-5 and -6 desaturases (encoded by FAD1 and FAD2), are reported to correlate with increased ALA and decreased EPA (with no impact on DHA) [57]. Thus, further comprehensive genetic studies are desired to understand the physiological modifier of the n-3 PUFA status and response to intervention.

## 4. Evidence of the n-3 PUFA-Mediated Cardiac Protection from Basic and Translational Research

Despite controversy regarding the benefits of n-3 PUFA for HF prevention in clinical trials, a significant amount of supportive evidence from basic and clinical translational research has been reported. Although such follow-up research tends to contain an inherent bias, by which positive data are likely to be reported, each study has well-explained pathophysiological mechanisms of preventive effects of PUFA against HF (Figure 2). In this section, we summarize the evidence of n-3 PUFA-mediated cardiac protection and HF prevention both from basic and translational research.

### 4.1. Anti-Inflammatory Effect of n-3 PUFA

It is well known that the inflammatory cytokines (e.g., interleukin-1β and TNF-α), which can reduce both systolic and diastolic cardiac function to advance cardiac remodeling through abnormal calcium handling in cardiomyocytes and enhancement of cardiac fibrosis via an activation of fibroblasts, are elevated both in the blood and local heart tissue of HF patients [58]. In addition, systemic inflammatory cytokines are considered one of the major causes of cachexia during HF [59,60]. Thus, the activation of local and/or systemic inflammatory reaction plays a pivotal role for the pathogenesis and progression of HF.

Classically, eicosanoids originated from n-3 PUFA are known to enhance the production of anti-inflammatory cytokines at the site of inflammation, whereas essential n-6 PUFAs, such as AA, are considered as substrates of pro-inflammatory eicosanoids, which promote vascular permeability, leucocyte infiltration and activation, and pro-inflammatory cytokine release [61]. The conventional interpretation is that the n-3 PUFAs antagonize the production and action of the inflammatory eicosanoid derived from AA metabolites. Indeed, in a pressure-overloaded HF rodent model (induced by aortic contraction surgery), the n-3 PUFA supplementation reduced serum TNF-α as well as pro-inflammatory eidosanoid (i.e., thromboxane B_2_), and prevented abnormal LV remodeling [62]. In addition, a number of human clinical investigations revealed that n-3 PUFA supplementation reduced the pro-inflammatory cytokines in terms of lower circulating levels of cytokines, e.g., TNF-α, IL-1, and IL-6 [24,32,63,64]. Basic studies also support the idea that n-3 PUFA would downregulate the pro-inflammatory pathways, such as NF-κB [65] and NLRP3 inflammasome [66], or upregulate anti-inflammatory intra-cellular signaling pathways, including peroxisome proliferator-activated receptor (PPAR) α/γ transcriptional activation [67]. Meanwhile, another study suggested the effect of low dose n-3 PUFA on NF-κB pathway activation in a cultured macrophage, potentially enhancing pro-inflammatory cytokines production [68]. The immunomodulatory activity of n-3 PUFAs has not been clearly explained yet and should be investigated further in detail.

Adiponectin, a peptide hormone released from adipose tissue, is known to show a cardio-protective effect through anti-inflammatory reaction. PPARγ-dependent adiponectin secretion is thought to correlate with n-3 PUFA-mediated cardiac protection, and the administration of n-3 PUFA increased the circulation of adiponectin in a dose-dependent manner both in animal [62,69] and human [70] studies.

The free FA specific membrane receptor family, a family of orphan G-protein coupled receptors (GPR), has been detected in the last 10 years. GPR120, also known as a free fatty acid receptor (FFR) 4, was detected as a receptor for n-3 PUFA, regulating downstream of intra-cellular signaling [71]. Eclov et al. revealed that the expression of FFR4 was markedly higher than other subtypes of FFRs in cardiomyocytes as well as fibroblasts isolated from mice [72]. In a rat heart study, a high-fat diet increased the expression level of FFR4 [73]. The involvement of FFR4 in the n-3 PUFA-mediated cardiac protection has been actively investigated in cardiac fibroblasts rather than cardiac myocytes. In a rodent pressure-overloaded HF model, EPA inhibited the transforming growth factor-beta 1 (TGF-β1) pro-fibrotic pathway via FFR signaling in cardiac fibroblasts and suppressed the entire cardiac fibrosis without requirement of EPA localization into the cellular membrane [72]. In addition, the activation of FFR4 by n-3 PUFA further stimulated endothelial nitric oxide synthase (eNOS) and promoted intracellular nitric oxide (NO), which leads to the suppression of TGF-β1 induced smad2/3 nuclear translocation and inhibition of pro-fibrotic gene transcription. It remains to be elucidated whether FFR4 is also expressed in human cardiomyocytes or cardiac fibroblasts.

In order to obtain enhanced benefits of n-3 PUFA-mediated cardiac protection, it would be better to identify the effective n-3 PUFA metabolites rather than to take a high dose of n-3 PUFA. Recently, the essential PUFA derived large class of cell signaling lipid mediators, named specialized pro-resolving mediators (SPMs; resolvins, protectins, lipoxins, and maresins) were discovered and indicated to exert the resolution of inflammatory reactions at nanomolar levels of concentration [74,75]. Halade et al., in a mouse coronary ligation model, reported that leukocytes, which immigrate from the splenic reservoir to the acutely infarcted myocardium, express lipoxygenases and abundant SPMs, predominantly derived from DHA to resolve the local acute inflammatory reaction [76]. Other rodent studies investigating the post-MI rodent heart revealed that the exogenous delivery of resolvins reduced infarction size [77] and excessive inflammation and fibrosis [78], leading to improved cardiac function [78]. Precursor of resolvins (i.e., 18-hydroxyeicosapentaenoic acid) administration in a pressure overload HF model mouse was also reported to ameliorate cardiac inflammation and fibrosis and preserve systolic function [79]. A recent clinical study revealed that plasma resolvin D1 levels were markedly decreased in patients with chronic HF compared with healthy subjects, suggesting a defect of resolvin biosynthesis in HF conditions [80]. To date, medicine purified or produced from SPMs has not been available or reached the stage of clinical application.

### 4.2. Effects on Myocardial Energy Metabolism and Mitochondrial Function of n-3 PUFA

Alteration of energetic substrate utilization in myocardium and modification of its intra-cellular signaling pathway also play important roles for the pathogenesis and progression of HF [81]. Previous animal studies revealed that the dietary n-3 PUFA intake altered the mitochondrial membrane phospholipid composition in cardiomyocytes, which led to a decrease in myocardial oxygen consumption without loss of ventricular power generation [82,83]. Furthermore, mitochondrial permeability transition pore (mPTP) opening, by which mitochondrial swelling and apoptotic cell death are activated, is suppressed by n3-PUFA (especially by DHA) supplementation [84,85,86]. Thus, it is considered that n-3 PUFA could protect the heart through improvement in the cardiac mitochondrial function as well as the efficiency of the ATP production [82,83,84,85,86].

In HF, lipotoxicity by excessive serum free FAs, which is caused by chronic adrenergic stimulation, is one of the critical pathophysiological mechanisms to exacerbate HF. Excessive free FA exposure to cardiomyocytes causes uncoupled mitochondrial respiration and reactive oxygen species (ROS) production [87], which thereby causes energy depletion and further impairs contraction in the failing heart [88]. Saturated fatty acids (SFA) (e.g., palmitate and stearate) are reported as the main components of serum free FAs [89]. Since unsaturated FAs, especially n-3 PUFA, are reported to act counter to the behavior of SFA, it is widely accepted that n-3 PUFA may mitigate SFA-induced lipotoxicity.

Mitochondria are dynamic organelles, which can continuously alter their morphology to maintain a number of cellular processes, such as cell cycle, immunity, apoptosis, and mitochondrial quality control. Mitochondrial dynamics play key roles for the pathophysiology of HF [90,91]. Specific fusion-related (e.g., mitofusin 1, mitofusin 2, and optic atrophy 1) and fission-related (e.g., dynamin related protein 1 (Drp1), mitochondrial fission 1 protein, and mitochondrial fission factor) proteins are involved in the regulation of mitochondrial dynamics [92]. We, in our previous investigation, reported that EPA activated phosphorylation of AMP-activated protein kinase (AMPK; a key enzyme for cellular energy homeostasis), altered mitochondrial morphology (relatively elongated mitochondria by suppression in Drp1) in myocardium, and thereby protected myocytes from SFA-induced cardiac lipotoxicity [93]. Despite the attractive pathophysiological findings of n-3 PUFA for cardiac protection against lipotoxicity, further in vivo animal and/or human investigations are warranted.

### 4.3. Anti-Arrhythmic Property by n-3 PUFA

Patients with HF are highly comorbid with arrhythmia, and the incidence of arrhythmia certainly worsens the rate of mortality of HF [94]. n-3 PUFAs are reported to reduce both atrial and ventricular arrhythmia, which may cause an improvement in the mortality of patients with HF. In the GISSI-HF trial, the reason for CVD risk reduction was mainly presumed to be due to anti-arrhythmic effects [10]. A number of basic experimental studies have revealed direct and/or indirect alteration in the electrophysiological behavior of plasma membrane ion channels of cardiomyocytes, such as the sodium, potassium and calcium channels, as well as the sodium-calcium exchanger [95,96]. In isolated mammalian cardiomyocytes (e.g., neonatal and adult rat ventricular myocytes), n-3 PUFA exhibited the inhibition of sodium current in a dose-dependent manner [97,98]. In addition, n-3 PUFA suppressed an intracellular Ca^2+^ wave, which was propagated from sarcoplasmic reticulum (SR) Ca^2+^ release by isoproterenol stimulation [99], suggesting a contribution of n-3 PUFA to suppression of arrhythmogenicity (i.e., myocardial triggered activity and abnormal automaticity) during HF.

Although the detailed mechanism remains unknown, n-3 PUFA may be able to improve the autonomic nervous system. An impaired autonomic tone is one risk-factor of a fatal arrhythmic event and sudden cardiac death in patients with dilated cardiomyopathy [100] and ischemic cardiomyopathy [101]. Some clinical studies revealed that n-3 PUFA supplementation improved heart rate variability (an index of autonomic tone) in patients with ischemic and non-ischemic cardiomyopathy [39,40,102,103]. Thus, n-3 PUFA may improve the mortality of HF patients through the autonomic nervous system and thereby suppression of anti-arrhythmogenicity.

### 4.4. Anti-Hypertensive Effect, Improvement of Vascular Endothelial Function, and Modulation of Autonomic Nervous System Activity by n-3 PUFA

It is well known that hypertension is a critical factor in developing HF. Previous investigations revealed the beneficial effects of fish oil on hypertension. Morris et. al. reported that the intake of fish oils reduced blood pressure (BP) by 3.0 mmHg in systole and by 1.5 mmHg in diastole (95% CI: systolic BP (4.5–1.5), diastolic BP (2.2–0.8)) [6]. Geleijnse et. al. also revealed that fish oils (mean intake 3.7 g/day) reduced BP by 2.1 mmHg in systole and 1.6 mmHg in diastole (95% CI: systolic BP (1.0–3.2), *p* < 0.01, diastolic BP (1.0–2.2), *p* < 0.01), and this anti-hypertensive effect of fish oil was obvious especially in the elderly and patients with hypertension [104]. The n-3 PUFA has been reported to release NO from vascular endothelial cells both in vivo [105] and in vitro [106]. In addition, DHA is reported to activate NO synthase and concentration of tetrahydrobiopterin in the central nervous system, which may increase local NO availability and exert tonic inhibition of central sympathetic outflow [107]. Thus, n-3 PUFA may suppress the progression of HF, not only through BP lowering, but also the correction of autonomic imbalance.

### 4.5. Anti-Thrombotic and Anti-Atherosclerotic Effects and Prevention of HF by n-3 PUFA

The anti-thrombotic and anti-atherosclerotic effects by n-3 PUFA can contribute to HF prevention through the risk reduction of ischemic heart disease [3,16]. The n-3 PUFA is reported to suppress the synthesis of platelet-derived thromboxane A2 (TXA2), which causes platelet aggregation and vasoconstriction [108], and to increase the plasminogen activator inhibitor-1 with reduction of fibrinogen [109]. Atherosclerotic plaque stabilization by n-3 PUFA has been reported. Matsumoto et al. reported that EPA significantly suppressed the development of atherosclerotic lesions in atherosclerosis-prone mice with reduced production of matrix metalloproteinases released by macrophages in a PPARα-dependent fashion [110]. RCTs of patients awaiting carotid endarterectomy also revealed that the supplementation of n-3 PUFA substantially increased tissue concentration of EPA and DHA, and decreased macrophage infiltration and thickened the fibrous cap in the human carotid artery [111].

## 5. Summary, Conclusions, and Future Direction

In this review, we summarized the current knowledge regarding the relationship between n-3 PUFA and HF from both clinical and basic studies. Although there remains conflicting results regarding the effect of n-3 PUFA on primary HF prevention, this probably originates from the variety of HF subtypes, reliability of FFQs, participants’ genetic background for n-3 PUFA metabolism, and contamination of fish with pollutants that potentially increase cardiovascular disease risk. Regarding secondary HF prevention, the GISSI-HF showed that n-3 PUFA intervention slightly but significantly suppressed the mortality and hospitalization of HF patients. Considering the results of other small-scale studies regarding ischemic and/or non-ischemic HFrEF using detailed cardiac function assessments by imaging modalities (e.g., echocardiography), as well as the measurement of several inflammatory markers, n-3 PUFA supplementation can reduce ventricular remodeling and myocardial fibrosis, and improve ventricular systolic and diastolic function in a dose-dependent fashion. The differences of the quality and quantity of n-3 PUFAs adopted in each clinical study, and the cause and severity of HF in the study population, as well as other baseline clinical background factors, including genetic variance in terms of n-3 PUFA bioavailability, may be responsible for the inconsistent results of clinical trials of secondary prevention for HF. Basic experimental data support our understanding of the mechanisms of n-3 PUFA cardiac protection against HF, including: (1) Anti-inflammatory properties originated from classical eicosanoid production, as well as newly discovered highly bioactive PUFA metabolites, to reduce cardiac remodeling caused by excessive interstitial fibrosis and whole body inflammatory reaction leading to cardiac cachexia; (2) alteration of metabolic conditions of cardiomyocyte probably accompanied by mitochondrial functional modification; (3) direct and/or indirect functional modification of cardiac ion channels to reduce susceptibility for fatal arrhythmia; (4) anti-hypertensive effects originated from improvement of vascular endothelial response; and (5) modulation of autonomic nervous system activity.

To ensure the benefits of n-3 PUFA for HF prevention, further investigation should aim to (1) titrate the ideal n-3 PUFA dose by monitoring blood and/or tissue PUFA concentrations, (2) define the bioavailability of n-3 PUFA that affords better CVD outcomes, and (3) understand the genetic variants that are physiological modifiers of the status of n-3 PUFA.

## Figures and Tables

**Figure 1 ijms-20-04025-f001:**
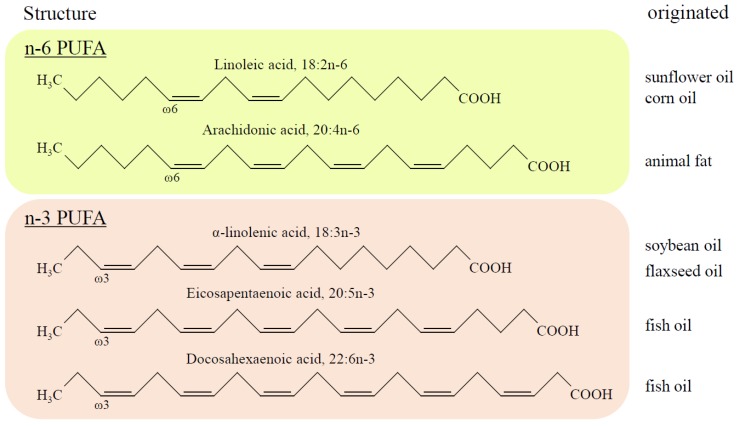
The structure of polyunsaturated fatty acids (PUFAs) and originated oils. The scheme presents the structures of PUFAs and their originated oils. The location of the first double bond from the methyl (-CH_3_) or omega (n-) end of the chain gives the name of the PUFA, i.e., the n-3 PUFAs have a double bond at the site of the third carbon, and the n-6 PUFAs have a bond at the sixth carbon from the methyl end of the chain.

**Figure 2 ijms-20-04025-f002:**
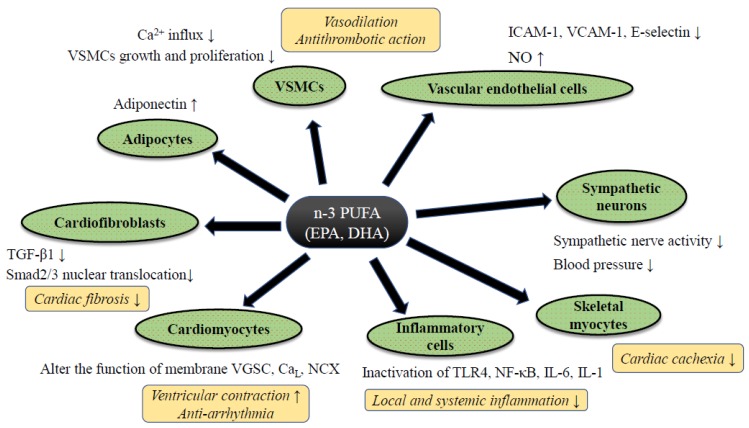
The putative mechanism of n-3 PUFA-mediated cardiac protection against heart failure. Abbreviations: Ca_L_; L type calcium channel, IL-6/-1; ICAM-1; intercellular adhesion molecule-1, Interleukin-6/-1, NCX: sodium calcium exchanger, NF-κB; nuclear factor-kappa B, NO; nitric oxide, TGF-β1; transforming growth factor-beta 1, TLR4; Toll-like receptor 4, VGSC; voltage-gated sodium channel, VCAM-1; vascular cell adhesion molecule-1, VSMCs; vascular smooth muscle cells. Abbreviations: ADMA: Asymmetric Dimethylarginine, DB: Double blind trial, ESLVD: End-systolic left ventricle diameters, Etv/Atv: Early rapid right ventricular filling/late right ventricular filling, FMD: Flow-mediated dilatation, LAEF: Left atrial ejection fraction, LVESVI: Left ventricular end-systolic volume indexed to body surface area, MC: Multi-center trial, MCP-1: Monocyte chemoattractant protein 1, NYHA: New York Heart Association class, OL: Open label trial, PC: Placebo-control trial, PRSP: Prospective trial, RDM: Randomized trial, SC: Single-center trial, ST2: Suppression of tumorigenicity 2, TDI: Tissue Doppler imaging. Downward allows represent the decrease or suppression, and upward allows the increase or enhancement.

**Table 1 ijms-20-04025-t001:** Primary prevention of heart failure (HF) clinical trials.

Study or Author/Reference	Year of Publication	Region	Follow Up (Years)	Study Population	HF Event	Incidence of HF
Cardiovascular Health study [8]	2005	USA	12	4.738 (male 42%, age > 65)	955	Boiled or baked fish intake was negatively associated
Woman’s Health Initiative Observational study [9]	2011	USA	10	84.493 (all female, age 50–79)	1.858	Boiled or baked fish intake was negatively associated
Physicians’ Health study [22]	2012	USA	14	18.968 (fish consumption analysis) 19.097 (dietary n-3 PUFA analysis, all male, age > 58.7)	695/703	Fish consumption greater than once per month was negatively associated
JACC study [23]	2008	Japan	12.7	57.972 (male 40%)	307	Fish and n3 PUFA intake were negatively associated
Rotterdam study [25]	2009	Nederland	11.4	5.299 (male 41%, age > 55)	669	Fish/n-3 PUFA intake was not associated
Levitan EB et. al. [26]	2009	Sweden	7	39.367 (all male, middle and old age)	597	Fish/n-3 PUFA intake was not associated
Levitan EB et. al. [27]	2010	Sweden	9	36.234 (all female, age 48–83)	651	Moderate consumption of fatty fish and n-3 PUFA were negatively associated

All trials were done with an observational cohort study. To date, no published randomized control trials (RCTs) have assessed the effect of dietary fish and n-3 PUFA intake on primary prevention of HF. Accordingly, the advisory form of the American Heart Association (AHA) has no recommendation of the n-3 PUFA intake for the purpose of HF primary prevention thus far [29].

**Table 2 ijms-20-04025-t002:** Clinical trials for secondary HF prevention.

Study or Author/Reference	Year of Publication	Study Design	Number of Patients	Region	n-3 PUFA	Baseline Patient Background	Follow Up	Outcomes	Interpretation
GISSI-HF [10]	2008	MC, RDM, DB, PC	3494; n-3 PUFA 3481; placebo	Italy	1 g/day	Mean age; 67 y, male 78%, NYHA; II 63%, III 34%, IV 3%, Mean EF; 33%	3.9 years	All-cause death or admission to hospital for cardiovascular reasons; HR 0.92 (99% CI 0.849–0.999)	n-3 PUFA can provide a small benefit for mortality and hospitalization
Zhao et. al. [31]	2009	MC, RDM, DB, PC	38; n-3 PUFA 37; placebo	China	2 g/day	Mean age; 73 y, male 73%, NYHA; II 37%, III 63%, Mean EF; 31%	3 months	Reduced in serum NT-proBNP (*p* < 0.001), TNF-α (*p* = 0.014), IL-6 (*p* = 0.003), and ICAM-1 (*p* = 0.023)	n-3 PUFA can reduce levels of plasma inflammatory markers and NT-proBNP
GISSI-HF (Echo sub-study) [30]	2010	MC, RDM, DB, PC	312; n-3 PUFA 296; placebo	Italy	1 g/day	Mean age; 65 y, male 84%, NYHA; II 77%, III 22%, IV 1%, Mean EF; 31%	3 years	Increased in LVEF (*p* = 0.005)	n-3 PUFA can provide a small advantage in terms of LV function
Nodari et. al. [24]	2011	SC, RDM, DB, PC	67; n-3 PUFA 66; placebo (olive oil)	Italy	5 g/day for 1mon 2 g/day for 11mon	Mean age; 62 y (18 to 75), NYHA; I 14%, II 86% Mean EF 36%	1 years	Increased LVEF and Peak VO_2_. Improved in exercise duration and NYHA. Reduced in Hospitalization. (all *p* < 0.001)	n-3 PUFA increased LV systolic function and functional capacity, and reduce HF hospitalizations
Mehra et. al. [32]	2006	SC, RDM, DB, PC	7; n-3 PUFA 7; placebo (corn oil)	USA	8 g/day	Mean age; 57 y, male 71%, NYHA; III 57%, IV 43%, Mean EF 17%	4.5 months	Decreased in TNF-α and IL-1	n-3 PUFA decreased TNF-α production in HF
Moertl et. al. [33]	2011	SC, RDM, DB, PC, 3-arm	14; n-3 PUFA (1g/d) 13; n-3 PUFA (4g/d) 16; placebo	Austria	1 g/day or 4 g/day	Mean age; 58 y, male 86%, NYHA; III 91%, IV 9%, Mean EF; 24%	3 months	Increased LVEF (4 g/day; +5%, 1 g/day; +3%). Reduced hs IL-6 by 2.3 pg/mL (*p* = 0.01 vs baseline)	n-3 PUFA dose dependently improved LVEF and decreased serum IL-6
Kojuri et. al. [34]	2013	SC, RDM, DB, PC	38; n-3 PUFA 32; placebo	Iran	2 g/day	Mean age; 57 y, male 60%, NYHA; II to III, Mean EF; 31%	6 months	Reduced late diastolic velocity index, Tei index and plasma BNP	n-3 PUFA slightly decreased plasma BNP levels and moderately improved ventricular diastolic function.
Kohashi et. al. [35]	2014	SC, OL, PRS	71; EPA 68; no EPA	Japan	EPA 1.8mg/day	Mean age; 70 y, male 86%, NYHA; II 91%, III 9%, Mean EF; 37.6%	1 year	Increased LVEF. Reduced MCP-1and ADMA. Suppressed cardiac death and HF readmission; HR 0.21 (95% CI 0.05–0.93)	EPA improved cardiac function and prognosis of HF
OMEGA-REMODEL [36]	2016	MC, RDM, DB, PC	180; n-3 PUFA 178; placebo (corn oil)	USA	4 g/day	Mean age; 59 y, male 80%, NYHA; I 91%, II 8%, III 1%, Mean EF 54%	6 months	Reduced LVESVI and non-infarction myocardial fibrosis and ST2	High dose n-3 PUFA reduced LV remodeling, myocardial fibrosis, and inflammatory biomarkers in patients with post AMI.
Chrysohoou et. al. [37]	2016	SC, RDM, OL, PRS	101; n-3 PUFA 95; without n-3 PUFA (no placebo)	Greece	1 g/day	Mean age; 63y, male 83%, NYHA; I-III, Median EF; 28%	6 months	Reduce ESLVD, LAEF, TDI Etv/Atv and BNP	n-3 PUFA improved LV diastolic function and decreased BNP levels
Oikonomou et. al. [38]	2019	SC, DB, PC, cross over	15 vs 16; n-3 PUFA/placebo (olive oil, cross-over with 6 weeks wash-out period)	Greece	2 g/day	Mean age; 67 y (18 to 80), NYHA; II 65%, III 35%, Mean EF 29%,	2 months	Increased LVEF Reduced global longitudinal strain, E/e’ ratio, hsCRP, ST2 levels, FMD % increase	n-3 PUFA improved inflammatory, fibrotic, and endothelial functional status as well as systolic and diastolic LV function.

Abbreviations: ADMA: Asymmetric Dimethylarginine, DB: Double blind trial, ESLVD: End-systolic left ventricle diameters, Etv/Atv: Early rapid right ventricular filling/late right ventricular filling, FMD: Flow-mediated dilatation, LAEF: Left atrial ejection fraction, LVESVI: Left ventricular end-systolic volume indexed to body surface area, MC: Multi-center trial, MCP-1: Monocyte chemoattractant protein 1, NYHA: New York Heart Association class, OL: Open label trial, PC: Placebo-control trial, PRSP: prospective trial, RDM: Randomized trial, SC: Single-center trial, ST2: Suppression of tumorigenicity 2, TDI: Tissue Doppler imaging.

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
