# Peer review of "Marine-Derived Omega-3 Polyunsaturated Fatty Acids and Heart Failure: Current Understanding for Basic to Clinical Relevance"

_ijms, 2019, doi:10.3390/ijms20164025_

Round 1

Reviewer 1 Report

The thought that increasing intake of polyunsaturated fatty acids (PUFA) may have beneficial effects on heart is attractive to both physicians and patients. However, the results from clinical studies remain controversial. The current review summarized the results of clinical trials in the scope of prevention of heart failure by n-3 PUFA and the potential molecular basis of n-3 PUFA. These knowledges may help us to better understand and make decisions on the usage of n-3 PUFA. There are some minor points need to be addressed.

1) Heart failure results from diverse pathological heart conditions, eg. coronary heart diseases, hypertension, etc. The different molecular basis may affect the performance of the supplementation of n-3 PUFA. Authors should further provide such information and the potential effects on the supplementation of n-3 PUFA.

2) In Table 1, "associate" in the column "Primary HF Prevention" should be changed to a word to clearly show the relationship (negatively/positively) with the occurrence of HF.

3) Relevant papers should be included in the manuscript, giving a clearer scope on this topic, eg. Wang C, et al. Nutrients. 2017 Jan; 9(1): 18.

4) Relevant RCT should be included in the manuscript, eg. [1] Radaelli A, et al. J. Am. Coll. Cardiol. 2006;48:1600–1606; [2] Nodari S, et al.  Cardiovasc. Drugs Ther. 2009;23:5–15.

Author Response

Response to Reviewer #1:

We appreciate the careful reading of our manuscript and constructive comments of reviewers, which has substantially helped to improve the quality of our manuscript. In the revised version of our manuscript, we tried to respond to all the criticisms and suggestions raised by the reviewers, as listed below. Reviewer’s comments are in bold, and the responses are written in plain text.

1) Heart failure results from diverse pathological heart conditions, eg. Coronary heart diseases, hypertension, etc. The different molecular basis may affect the performance of the supplementation of n-3 PUFA. Authors should furthe provide such information and the potential effects on the supplementation of n-3 PUFA.

This comment from reviewer is right. Because heart failure happens divers pathological results, n-3 PUFA may have pleotrophic effects which corresponding to each pathophysiologies. According to this reviewer’s suggestion, we added texts regarding the prevention coronary atherosclerotid disease by n-3 PUFA as listed below.

Added in Page 9, line 8:

Anti-thrombotic and anti-atherosclerotic effects and prevention of HF by n-3 PUFA

The anti-thrombotic and anti-atherosclerotic effects by n-3 PUFA can contribute to HF prevention through the risk reduction of ischemic heart disease [3, 16]. n-3 PUFA is reported to suppress the synthesis of platelet-derived thromboxane A2 (TXA2), which causes platelet aggregation and vasoconstriction [108], and to increase the plasminogen activator inhibitor-1 with reduction of fibrinogen [109]. Atherosclerotic plaque stabilization by n-3 PUFA has been reported. Matsumoto et al. reported that EPA significantly suppressed the development of atherosclerotic lesions in atherosclerosis-prone mice with reduced production of matrix metalloproteinases released by macrophages in a PPARa-dependent fashion [110]. RCTs of patients awaiting carotid endarterectomy also revealed that the supplementation of n-3 PUFA substantially increased tissue concentration of EPA and DHA, and decreased macrophage infiltration and thickened the fibrous cap in the human carotid artery [111].

2) In Table 1, "associate" in the column "Primary HF Prevention" should be changed to a word to clearly show the relationship (negatively/positively) with the occurrence of HF.

We appreciate this reviewer’s suggestion. We have corrected the description in the column of Table 1.

3) Relevant papers should be included in the manuscript, giving a clearer

scope on this topic, eg. Wang C, et al. Nutrients. 2017 Jan; 9(1): 18.

This suggestion is reasonable. We added the text regarding this manuscript as described below.

Added in Page 8, line 11,

Meanwhile, another study suggested the effect of low dose n-3 PUFA on NF-κB pathway activation in a cultured macrophage, potentially enhancing pro-inflammatory cytokines production [68]. The immunomodulatory activity of n-3 PUFAs has not been clearly explained yet and should be investigated further in detail.

4) Relevant RCT should be included in the manuscript, eg. [1] Radaelli A, et al.

Am. Coll. Cardiol. 2006;48:1600–1606; [2] Nodari S, et al. Cardiovasc.

Drugs Ther. 2009;23:5–15.

We appreciate this reviewer’s reasonable suggestion. We added the following description in the revised manuscript.

Added in Page 6, line 15,

Cardiovascular event reduction in the GISSI-HF trial was mainly obtained by anti-arrhythmic effects [10], and small-scale clinical trials also supported anti-arrhythmic effects by n-3 PUFA in patients with HF [39, 40]. These data suggest the anti-arrhythmic effects of n-3 PUFA may contribute to CVD events in patients with HF.

Reviewer 2 Report

In this review article, the authors attempt to summarize and evaluate the use of n3-PUFA as a therapeutic agent for heart failure.  The breadth and depth of the review of clinical trials and basic/translational science suggesting mechanisms is excellent.  However, I have three major concerns: 1. The article will need to be thoroughly re-written with the help of an English language editor; 2. The article is not entirely balanced. It does not give sufficient weight to the arguments against n3-PUFA; 3. The article does not clearly lay out for the reader what further study is needed to advance the field.

Abstract:

The goal set out in the abstract “to achieve maximum benefits of n3-PUFA” should be modified to reflect a more balanced approach. The goal is to most clearly lay out the pros and cons of using this therapeutic strategy.

Introduction:

The phrase “steeled the limelight” contains a misspelling and is overly colloquial for such an article.

Structure and Metabolism of PUFA

This section would benefit greatly from a figure showing the chemical structure of n3-PUFA and a diagram describing its biological interactions

Clinical evidences of n-3 PUFA on heart failure

This section requires a clear summary paragraph at the end. The authors appear to attempt to do so under the following heading “Considerable points of clinical trials to interpret the heterogeneous CVD outcomes”, but this section is quite ineffective as it is a sentence that does not tell the reader much. The summary paragraph should make it clear what the authors think is known, what is not known, and what could be done in terms of future trials to settle any remaining controversy.

Effective dose of PUFA

The authors fail to point out that this is a major weakness of n3-PUFA. A medication for which the effective dose is unclear is difficult to use therapeutically.

Evidences of the n-3 PUFA-mediated cardiac protection from basic and translational research

I think in this section it is important to mention the biases inherent in this type of research. Studies of this kind are highly likely to report “positive” findings and very unlikely to report “negative” findings, and thus they collectively can present a picture that suggests very robust effects of n3-PUFA when in fact evidence might be mixed.

Summary, conclusion, and future direction

The second paragraph needs to be considerably improved. The authors need to clearly lay out for readers what further studies need to be done in order to clarify the putative role of n3-PUFA in therapy for heart failure.

Author Response

Response to Reviewer #2:

We appreciate the careful reading of our manuscript and constructive comments of reviewers, which has substantially helped to improve the quality of our manuscript. In the revised version of our manuscript, we tried to respond to all the criticisms and suggestions raised by the reviewers, as listed below. Reviewer’s comments are in bold, and the responses are written in plain text.

Abstract:

The goal set out in the abstract “to achieve maximum benefits of n3-PUFA” should be modified to reflect a more balanced approach. The goal is to most clearly lay out the pros and cons of using this therapeutic strategy.

 This comment from reviewer is right. According to this reviewer’s suggestion, we corrected texts in abstract as listed below.

Corrected in abstract (page 1, line 15)

To clarify the pros and cons of n-3 PUFA on HF, we summarized recent evidence regarding the beneficial effects of n-3 PUFA on HF both from clinical and basic studies.

Also, as this reviewer pointed below, we should mention that it remains many critical issue for the clinical use of PUFAs to HF. Thus the beginning of abstract was corrected as listed below.

Corrected in abstract (page 1, line 9)

Although it remains controversial, omega 3 polyunsaturated fatty acids (n-3 PUFAs), such as eicosapentaenoic acid and docosahexaenoic acid, have been widely recognized to have benefits for HF.

Introduction:

The phrase “steeled the limelight” contains a misspelling and is overly colloquial for such an article.

We appreciated this reviewers comment. According to this reviewer’s suggestion, we corrected texts in abstract as listed below.

Corrected in introduction, (page 2, line 5)

However, a recent large-scale clinical trial (REDUCE-IT; Reduction of Cardiovascular Events with Icosapent Ethyl–Intervention Trial) showed that a relatively high amount of n-3 PUFA supplementation (4 g/day) significantly reduced CVD events in patients with elevated serum triglyceride [16], and the beneficial effect of n-3 PUFAs intervention against CVD has been re-recognized.

Structure and Metabolism of PUFA:

This section would benefit greatly from a figure showing the chemical structure of n3-PUFA and a diagram describing its biological interactions.

According to this reviewer’s suggestion, we added a new figure (figure 1).

Clinical evidences of n-3 PUFA on heart failure:

This section requires a clear summary paragraph at the end. The authors appear to attempt to do so under the following heading “Considerable points of clinical trials to interpret the heterogeneous CVD outcomes”, but this section is quite ineffective as it is a sentence that does not tell the reader much. The summary paragraph should make it clear what the authors think is known, what is not known, and what could be done in terms of future trials to settle any remaining controversy.

We appreciated this constructive comment from reviewer. According to this reviewer’s suggestion, we corrected texts and added clear summary paragraph in the bigging of this section as listed below.

Added in page 2, line 30

HF is a condition in which the heart is unable to pump enough blood to meet demand. Because HF is a multifactorial syndrome, its mortalities and progressions differ from their underling etiologies, such as IHD, valvular heart disease, hypertension, arrhythmia, cardiac myopathies, congenital heart disease, endocrine and metabolic diseases, infection, and certain drugs. Several prior prospective observational studies and randomized control trials proved that consumption of fish or fish oil containing n-3 PUFA decreased IHD mortality (e.g., myocardial infarction (MI) and sudden cardiac death) in patients with or without pre-diagnosed CVD [2, 19, 20]. In addition, recent studies have shown that fish and/or n-3 PUFA intake also prevented new onset of HF and rehospitalization [9, 10, 21-24]. However, conflicting results were reported from other groups [25-27], meaning that the n-3 PUFA-mediated effects on HF are once again viewed as uncertain. In this section, we summarize existing clinical evidence regarding primary and secondary prevention of HF, and discuss the considerable points which may cause such heterogenous results.

In addition, the part is surely tedious and confusing to the readers. In the revised manuscript we first summarized the point in the biggining of section and shortened the text for clarify. Also, because of the suggestion from the English editing service, we have changed the heading title “Considerable points of clinical trials to interpret the heterogeneous CVD outcomes” to “Key points which can affect heterogeneous CVD outcomes”.

Corrected in page 6, line 34

Key points which can affect heterogeneous CVD outcomes

As mentioned above, heterogeneous CVD outcomes have been reported from PUFA clinical trials. These heterogenous outcomes in CVD may be derived from differences in the quality and quantity of n-3 PUFAs adopted in each clinical study, and other baseline clinical background factors, including genetic variance in terms of n-3 PUFA bioavailability. In this section we discuss the key points which can affect the results of PUFA clinical trials.

Heterogeneity in blood and tissue PUFA concentration: Although a previous meta-analysis reported no statistically significant relationship between n-3 PUFA consumption and CVD mortality, the variability in individual serum n-3 PUFA levels was not taken into consideration in the study [43]. Superko et al. reported that fixed-dose n-3 PUFA administration exhibited significant individual variation in blood n-3 PUFA levels [44], whereas, in general, dietary intake of n-3 PUFA is closely associated with its blood levels [45, 46]. It is well known that the risk reduction of CVD depends on n-3 PUFA blood concentration [44], and an inverse correlation with the blood n-3 PUFA profiles has been reported in IHD [47, 48], arrhythmia [49], peripheral artery disease [50], and HF [21, 51]. Thus, heterogenous tissue and/or blood concentration in each patient (even for the same dose intake) may affect heterogenous CVD outcomes.

Effective dose and bioavailability of PUFA: Currently, few studies have provided an ideal n-3 PUFA dose, which can sufficiently correct the tissue PUFA concentration, and which may account for the heterogeneity of clinical outcomes. This unavailability of an effective dose of PUFA intake for HF prevention decreases the clinical evidence level. Thus, in order to use PUFA for HF prevention, we should first explore titration of the ideal n-3 PUFA dose by monitoring blood and/or tissue PUFA concentrations. In addition, the bioavailability of n-3 PUFA can be altered by its chemical form. In triglyceride or phospholipid forms (as usually found in fish oil), n-3 PUFA is mainly hydrolyzed by colipase-dependent pancreatic lipase. Meanwhile the bioavailability of the ethyl-ester form of n-3 PUFA is highly dependent on the presence of dietary fat, which can stimulate the release of bile salts into the small intestinal tract, because n-3 PUFA ethyl-esters requires digestion with bile salt-dependent lipase [52, 53]. Currently, the US Food and Drug Administration approves two ethyl-ester (Lovaza®, GlaxoSmithKline (EPA/DHA ethyl-esters) and Vascepa®, Amarin Pharmaceuticals (EPA ethyl-esters)) and one free fatty acid (Epanova®, AstraZeneca (EPA/DHA free fatty acids)) forms of n-3 PUFA supplement. Thus, the different chemical forms and bioavailability of n-3 PUFA can affect the clinical outcome through absorption via the intestine. Regarding bioavailability, the free forms of n-3 PUFA seem to provide better CVD outcomes than the ethyl-ester forms [54, 55]. Therefore, further clinical trials to investigate differences of bioavailability are required.

Genetic variant in PUFA metabolism: It is likely that genetic polymorphism affects individual taste-perception, which decisively determines the intake of food containing n-3 PUFA (e.g., oily fish). Prior familial aggregation analysis indicated the strong heritability of FA composition in the Red blood cells (RBC) membrane (up to 40–70%) [56]. Endogenous synthesis of n-3 PUFAs from plant-derived ALA appears mainly in the liver via desaturation and elongation reactions, even though the estimated conversion rate is low: 5% to 10% for EPA, and 2% to 5% for DHA [17]. Genetic variants of the key enzymes in this pathway, including delta-5 and -6 desaturases (encoded by FAD1 and FAD2), are reported to correlate with increased ALA and decreased EPA (with no impact on DHA) [57]. Thus, further comprehensive genetic studies are desired to understand the physiological modifier of n-3 PUFA status and response to intervention.

 Effective dose of PUFA:

The authors fail to point out that this is a major weakness of n3-PUFA. A medication for which the effective dose is unclear is difficult to use therapeutically.

 We really appreciated this reviewer comment. We agreed this is most critical point of PUFA medication. We mentioned this issue in the revised manuscripy as listed below.

Corrected in Page 7, line 11.

This unavailability of an effective dose of PUFA intake for HF prevention decreases the clinical evidence level. Thus, in order to use PUFA for HF prevention, we should first explore titration of the ideal n-3 PUFA dose by monitoring blood and/or tissue PUFA concentrations. In addition, the bioavailability of n-3 PUFA can be altered by its chemical form.

Evidences of the n-3 PUFA-mediated cardiac protection from basic and translational research:

I think in this section it is important to mention the biases inherent in this type of research. Studies of this kind are highly likely to report “positive” findings and very unlikely to report “negative” findings, and thus they collectively can present a picture that suggests very robust effects of n3-PUFA when in fact evidence might be mixed.

 We appreciated this constructive comment from reviewer. According to this reviewer’s suggestion, we corrected texts and clearly mentioned about the possibility of inherent bias as dyscribed below.

Added in Page 7, line 28.

Despite controversy regarding the benefits of n-3 PUFA for HF prevention in clinical trials, a significant amount of supportive evidence from basic and clinical translational research has been reported. Although such follow-up research tends to contain an inherent bias, by which positive data are likely to be reported, each study has well-explained pathophysiological mechanisms of preventive effects of PUFA against HF (Figure 2). In this section, we summarize the evidence of n-3 PUFA-mediated cardiac protection and HF prevention both from basic and translational research.

 Summary, conclusion, and future direction:

The second paragraph needs to be considerably improved. The authors need to clearly lay out for readers what further studies need to be done in order to clarify the putative role of n3-PUFA in therapy for heart failure.

 According to this reviewer’s suggestion, we corrected the second paragraph as described below.

Corrected in Page 11, line 1.

To ensure the benefits of n-3 PUFA for HF prevention, further investigation should aim to (1) titrate the ideal n-3 PUFA dose by monitoring blood and/or tissue PUFA concentrations, (2) define the bioavailability of n-3 PUFA that affords better CVD outcomes, and (3) understand the genetic variants that are physiological modifiers of the status of n-3 PUFA.

Round 2

Reviewer 1 Report

The current review thoroughly summarized the clinical findings on the supplementation of n-3 PUFA in patients with HF. It may help us to better understand or evaluate the use of n-3 PUFA in clinical practice. The current manuscript can be considered for the publication.

Reviewer 2 Report

My original critique has been well addressed and I am satisfied with this version of the manuscript.